# The Global Prevalence and Associated Factors of Burnout among Emergency Department Healthcare Workers and the Impact of the COVID-19 Pandemic: A Systematic Review and Meta-Analysis

**DOI:** 10.3390/healthcare11152220

**Published:** 2023-08-07

**Authors:** Ahmed Ramdan M. Alanazy, Abdullah Alruwaili

**Affiliations:** 1Emergency Medical Services Department, College of Applied Medical Sciences, King Saud Bin Abdulaziz University for Health Sciences, Al Ahsa 36428, Saudi Arabia; ruwailia1@yahoo.com; 2King Abdullah International Medical Research Center, Al Ahsa 11481, Saudi Arabia; 3Ministry of National Guard—Health Affairs, Al Ahsa 11426, Saudi Arabia

**Keywords:** burnout, emotional exhaustion, personal accomplishment, depersonalization, emergency medicine

## Abstract

Background/Aim: Emergency medicine (EM) settings are very stressful, given the high workload, intense working environment, and prolonged working time. In turn, the rate of burnout and its three domains have been increasingly reported among healthcare workers (HCWs). Therefore, we conducted this meta-analysis to determine the prevalence and risk factors of burnout among EM HCWs. Methods: Six databases were searched in February 2023, yielding 29 articles (16,619 EM HCWs) reporting burnout or its three domains (emotional exhaustion “EE”, depersonalization “DP”, and personal accomplishment “PA”). The primary outcome was the prevalence of burnout and its domains, while secondary outcomes included the risk factors of high burnout, EE, DP, or low PA. Burnout rates were pooled across studies using STATA software. The prevalence was measured using the pooled effect size (ES), and the random-effects model was used when heterogeneity was encountered; otherwise, the fixed-effects model was used. Results: The prevalence of overall burnout was high (43%), with 35% of EM HCWs having a high risk of burnout. Meanwhile, 39%, 43%, and 36% of EM workers reported having high levels of EE and DP and low levels of PA, respectively. Country-specific changes in the rate of burnout were observed. The rate of high burnout, high EE, high DP, and low PA was higher during the COVID-19 pandemic as compared to the pre-pandemic period. The type of profession (nurses, physicians, residents, etc.) played a significant role in modifying the rate of burnout and its domains. However, gender was not a significant determinant of high burnout or its domains among EM workers. Conclusions: Burnout is a prevalent problem in emergency medicine practice, affecting all workers. As residents progress through their training years, their likelihood of experiencing burnout intensifies. Nurses are most affected by this problem, followed by physicians. Country-associated differences in burnout and its domains are evident.

## 1. Introduction

Emergency medicine (EM) training and clinical practice are very stressful ventures [1]. Time is usually a critical consideration, and emergency physicians are expected to be well-versed in assessing and caring for surgical, medical, and trauma situations. It is sometimes considered “a young person’s specialty” due to the high intensity of the working environment. EM physicians are required to assess patients in a more constrained time window and make judgments under pressure, unlike departments such as outpatient. They are sometimes required to make crucial decisions with insufficient information, for instance, when the patient is unconscious [2]. As a result of these challenges, emergency services are the most hectic of the services offered in the hospital. This puts EM healthcare workers at risk of developing “burnout syndrome”.

Burnout refers to a three-dimensional response to long-term emotional as well as interpersonal pressures at work. The three dimensions of burnout are depersonalization (DP), emotional exhaustion (EE), and a decreased feeling of personal accomplishment (PA). Maslach went on to summarize the triad of burnout as “exhaustion”, “cynicism”, and “inefficacy”, providing definitions that were more specific and aligned well with her measuring tool [3]. Those with high exhaustion scores feel overworked and their mental and physical resources depleted. People who score high on cynicism appear colder or more detached than regular people who are “coping”. Those who lack confidence or believe they have had little work success have a high efficacy score (decreased feeling of personal accomplishment) [4].

These definitions were used by Maslach to create the Maslach Burnout Inventory (MBI), the most widely used evaluation tool for detecting burnout [3]. This tool has 22 questions covering all three categories and gives you a score for each. The degree of burnout in a particular dimension is evaluated by the score. The higher the score, the higher the degree of burnout. It describes a range with higher scores corresponding to more severe symptoms and consequences rather than a definitive cut-off score for burnout as a diagnostic. The MBI is adjusted and condensed for specific populations and ease of use [3]. For instance, Maslach Burnout Inventory-Human Services Survey for Medical Professionals (MBI-HSS) is used in the medical setting [3].

Generally, burnout has been observed to occur in 18–80% of residents of different specialties, while EM healthcare workers’ rate ranges from 32 to 60% [5]. Burnout has numerous detrimental effects on healthcare workers’ personal and professional life. In the professional domain, it leads to the delivery of poor services and consequently reduced patient satisfaction, more medical errors, higher physician turnover, and decreased productivity. On the physician’s side, it may lead to suicide ideation, sadness, addiction, or even physical illness [6]. While burnout is widespread among EM staff and is linked to various adverse outcomes, not all EM workers experience it. In the same presumably challenging situation, some of the members will thrive. The coping style is one significant aspect that may influence the likelihood of developing burnout [7].

Little is known about the exact burden of burnout in EM settings, and previous reports indicate a wide variability in the reported rates that might be attributable to the profession type, pandemic status, year of training, and other factors [7,8,9,10,11,12,13]. Therefore, we are conducting this research to systematically analyze the available literature to determine whether or not burnout among EM healthcare workers exceeds the rare event assumption (of ≤5%), as well as to determine the key factors that affect the prevalence of this problem statement in EM settings.

## 2. Materials and Methods

### 2.1. Literature Search

This systematic review was performed per the Preferred Reporting Items for Systematic Reviews and Meta-Analyses (PRISMA). To obtain the relevant studies, a systematic search was conducted on major e-databases, including PubMed, Scopus, Web of Science, PsycINFO, and EMBASE. We retrieved records that were published from inception until February 2023. Additionally, we searched and retrieved the first 200 studies from Google Scholar, as a source of Grey Literature, as per recent recommendations [14]. Articles were selected based on their relevance to the topic. All studies relevant to burnout in emergency medical services were included in this research. Keywords used to perform the search in the online databases were (burnout OR “emotional exhaustion” OR depersonalization OR “personal accomplishment”) AND (“emergency medicine” OR “emergency healthcare” OR “emergency physicians” OR “emergency nurses” OR “emergency care” OR “emergency medical technicians”). The detailed search strategy is provided in Appendix A. A manual search process was carried out where the references of finally included studies were screened, and their titles were inserted into PubMed to look for relevant articles using the “similar articles” function [15].

### 2.2. Selection Criteria

After selecting the initial cross-sectional studies, their titles and abstracts were scanned to check whether they were relevant to the research topic. The studies were further examined to determine whether or not they met the predetermined eligibility criteria. Relevant studies were included if they fulfilled all of the following criteria: (a) cross-sectional in design, (b) including EM healthcare workers (including but not limited to physicians, nurses, administration, paramedics, etc.), (c) reporting the rate of burnout or one of its domains (emotional exhaustion (EE), depersonalization (DP), or personal accomplishment (PA)), (d) measured burnout using the Maslach Burnout Inventory-Health Service Survey (MBI-HSS) scale [16], and (e) published in the English language. On the other hand, studies were excluded if they included: (a) duplicated studies, (b) irrelevant populations, outcomes, or burnout measures, or (c) secondary research (i.e., reviews, commentaries, editorials, guidelines, etc.). The authors thoroughly read and evaluated all of the studies that met the predetermined inclusion criteria for this systematic review to determine their implications and relevance to the research. Consultation or consensus with the senior author addressed any conflicts or discrepancies.

### 2.3. Data Extraction and Quality Assessment

The data extraction process was conducted after an initial pilot testing to determine the key variables to be included in the data extraction sheet. The sheet was then designed using Excel Software, and it was composed of three parts. The first part included publications (i.e., year of publication, authors’ names, country, and study design) and included populations’ characteristics (i.e., sample size, profession level of EM workers, age, and gender). The second part included the main outcomes: burnout, EE, DP, and PA. The third part included the variables that will be analyzed as subgroups, including severity, profession type (nurses, physicians, etc.), training years (postgraduate year “PGY”), COVID-19 pandemic timing (pre vs. post), and gender (male vs. female).

For each study, the risk of bias was determined using the Newcastle Ottawa Scale for cross-sectional studies [17]. This scale assesses the quality of cross-sectional studies across three domains: selection (maximum five points), comparability (maximum two points), and outcome (maximum 3 points). Both the data extraction and risk of bias assessment processes were done by at least two authors, and any contradictions were solved by consultation with the senior author.

### 2.4. Data Synthesis

All statistical analyses were performed using STATA Software (version 17). The prevalence of burnout, along with its subscales, was measured using the metaprop command [18]. The cimethod (score) method was used to pool the effect size (ES) and the corresponding 95% confidence interval (CI) across studies. The fixed-effects model was selected when statistical heterogeneity was not significant. On the contrary, the random-effects model was chosen. Statistical heterogeneity was measured by the *I*^2^ statistic and its associated *p*-value. A *p*-value of more than 0.05 indicated significant heterogeneity. In the latter, a leave-one-out sensitivity analysis was performed to determine whether or not the reported estimate was mainly driven by a particular study. Subgroup analyses were carried out based on the participating country, the COVID-19 pandemic timing (before vs. during), the profession type of EM workers (physician vs. nurse vs. A/T vs. paramedic) and year of training, and gender (male vs. female).

## 3. Results

### 3.1. Search Results

A schematic diagram of the database search and screening processes is presented in Figure 1. The electronic search resulted in 2051 studies, all of which were inserted into EndNote Software for duplicate removal, where 361 were automatically removed. Then, the remaining articles were exported into an Excel sheet for screening. In this stage, 71 additional duplicated articles were removed, resulting in 1619 articles eligible for title and abstract screening, out of which 1582 studies were excluded based on our predetermined eligibility criteria. The full text of 37 articles was screened, and only 29 studies met our criteria [5,8,9,10,11,12,13,19,20,21,22,23,24,25,26,27,28,29,30,31,32,33,34,35,36,37,38,39,40]. Meanwhile, eight studies were excluded either because burnout was not reported (*n* = 6) or because the authors failed to classify the type of included healthcare worker (*n* = 2). Of note, the manual search yielded no additional results.

### 3.2. Characteristics of Included Studies

The baseline characteristics of included reports are presented in Table 1. All of the included studies were cross-sectional in design, the majority of which were conducted in the United States of America (USA) (*n* = 9), followed by Turkey (*n* = 3), Canada (*n* = 2), France (*n* = 2), Iran (*n* = 2), and Singapore (*n* = 2). A total of 16,619 EM healthcare workers were included, and most of whom were males (10,026/16,619; 60.32%). Two studies included ambulance crew, 16 studies included only EM physicians, and the remaining 11 studies included mixed EM healthcare workers (physicians “attendings, residents, or fellows”, registered nurses, paramedics, and/or administrative and technical (A/T) staff. Meanwhile, ten studies included resident physicians only. The mean age of included population ranged from 31.2 to 44 years of age. Noteworthy, more than one-third (11/30 studies) did not report data regarding age.

### 3.3. Quality of Included Studies

The overall quality of included cross-sectional studies was graded as fair in 5 studies, poor in 7 studies, and good in 17 studies. The main reason why some studies were graded as poor was the comparability domain due to the lack of a comparator group in the majority of included cross-sectional studies (Table 2).

### 3.4. Meta-Analysis of Prevalence

#### 3.4.1. Burnout

Eleven studies reported the rate of burnout in EM healthcare workers, and the meta-analysis revealed a high prevalence rate of 43% [95% CI: 32–54%; *I*^2^ = 0]. The leave-one-out sensitivity analysis revealed no significant change in the reported rate. According to severity [Figure 2], the majority of EM healthcare workers had a low risk of burnout [ES = 46%; 95% CI: 25–68%], while 35% [95% CI: 17–56%] of them had a high risk of burnout. Meanwhile, those with evident burnout at the time of patient recruitment constituted 7% [95% CI: 4–12%] of EM workers.

#### 3.4.2. EE

Twenty-nine studies reported the severity of EE among EM workers. The meta-analysis revealed that EM workers had a higher prevalence of high EE [22 studies, ES = 39%; 95% CI: 28–49%], while less than one-quarter of them had moderate EE [15 studies, ES = 23%; 95% CI: 20–28%] [Appendix A].

#### 3.4.3. DP

Twenty-nine studies reported the severity of DP among EM workers. The meta-analysis revealed that EM workers had a higher prevalence of high DP [21 studies, ES = 43%; 95% CI: 36–50%], while less than one-quarter of them had moderate DP [15 studies, ES = 25%; 95% CI: 18–32%] [Appendix A].

#### 3.4.4. PA

Twenty-nine studies reported the severity of PA among EM workers. The meta-analysis revealed that EM workers had a higher prevalence of low PA [20 studies, ES = 36%; 95% CI: 28–44%], while less than one-quarter of them had moderate PA [15 studies, ES = 28%; 95% CI: 22–34%] [Appendix A].

### 3.5. Subgroup Analysis of High Burnout

#### 3.5.1. Country

Singapore showed the highest rate of burnout [ES = 68%; 95% CI: 64–73%] among EM workers, followed by the US [ES = 43%; 95% CI: 25–63%] and France [ES = 30%; 26–34%], respectively [Appendix A].

#### 3.5.2. COVID-19 Pandemic

A significant difference in the prevalence of burnout was noted based on the timing of the COVID-19 pandemic. EM workers had drastically higher rates of burnout during the pandemic [ES = 58%; 95% CI: 35–80%] as compared to the pre-pandemic time [ES = 37%; 95% CI: 26–49%] [Figure 3].

#### 3.5.3. Profession and Year of Training

Studies including only EM physicians [ES = 51%; 95% CI: 28–74%] had a higher prevalence rate of burnout as compared to studies including mixed EM healthcare workers [ES = 38%; 95% CI: 22–55%] [Figure 4]. Only one study reported burnout in ambulance crew, revealing a rate of 29% [95% CI: 21–39%].

Noteworthy, among studies including mixed EM workers, burnout was highest among nurses [ES = 47%; 95% CI: 42–52%], followed by paramedics [ES = 41%; 95% CI: 19–65%] and physicians [ES = 34%; 95% CI: 0–66%], respectively. Meanwhile, the A/T staff had the lowest prevalence of burnout, accounting for 11% of the population [95% CI: 7–16%] [Appendix A].

Three studies reported the rate of burnout among EM residents. In the meta-analysis, we noted a steady progressive increase in the prevalence of burnout as residents advanced through their training from PGY-1 [ES = 45%; 95% CI: 11–82%] to PGY-4 [ES = 63%; 95% CI: 25–93%] [Appendix A].

#### 3.5.4. Gender

Five studies compared the occurrence of burnout between males and females. The meta-analysis revealed no significant difference in the risk of burnout based on gender [logOR = −0.04; 95% CI: −0.17: 0.09; *I*^2^ = 57.31%, *p* = 0.053] [Figure 5].

### 3.6. Subgroup Analysis of High EE

#### 3.6.1. Country

A total of 22 studies were included in this meta-analysis. Turkey showed the highest prevalence of high EE among EM workers [ES = 50%; 95% CI: 48–54%], followed by the US [ES = 42%; 95% CI: 32–52%]. Meanwhile, France had the lowest prevalence of all [ES = 15%; 12–18%] [Appendix A].

#### 3.6.2. COVID-19 Pandemic

In this meta-analysis, 18 studies were conducted during the pre-pandemic period, while 4 studies were conducted during the COVID-19 pandemic. The prevalence of high EE almost doubled during the pandemic [ES = 58%; 95% CI: 44–72%] as compared to the pre-pandemic period [ES = 34%; 95% CI: 22–47%] [Figure 6].

#### 3.6.3. Profession and Year of Training

Studies including only EM physicians [ES = 44%; 95% CI: 32–55%] had the highest prevalence of high EE as compared to studies including mixed EM workers [8 studies, ES = 38%; 95% CI: 25–52%] or ambulance crew [2 studies, ES = 8%; 95% CI: 7–10%] [Appendix A].

Noteworthy, among studies including mixed EM workers, high EE was highest among nurses [ES = 51%; 95% CI: 16–85%] followed by physicians [ES = 42%; 95% CI: 21–64%] and A/T staff [ES = 34%; 95% CI: 6–71%], respectively. Meanwhile, paramedics had the lowest prevalence of high EE, accounting for 15% of the population [95% CI: 12–19%] [Appendix A].

Two studies reported the rate of high EE among EM residents. The meta-analysis revealed an increase during PGY-2 [ES = 53%; 95% CI: 49–58%], which was similar to that of PGY-3 [ES = 48%; 95% CI: 44–52%] as compared to PGY-1 [ES = 39%; 95% CI: 35–43%] [Figure 7].

#### 3.6.4. Gender

Two studies reported the difference in high EE based on gender. The meta-analysis revealed no significant difference in the risk of developing high EE between males and females [logOR = 0.47; 95% CI: −0.79: 1.73; *I*^2^ = 87.39%, *p* = 0.001] [Appendix A].

### 3.7. Subgroup Analysis of High DP

#### 3.7.1. Country

A total of 22 studies were included in this meta-analysis. The US showed the highest prevalence of high DP among EM workers [ES = 53%; 95% CI: 34–76%], followed by Turkey [ES = 50%; 95% CI: 48–52%]. Meanwhile, Iran had the lowest prevalence of all [ES = 29%; 23–35%] [Appendix A].

#### 3.7.2. COVID-19 Pandemic

In this meta-analysis, 18 studies were conducted during the pre-pandemic period, while 4 studies were conducted during the COVID-19 pandemic. The prevalence of high DP was higher during the pandemic [ES = 51%; 95% CI: 45–57%] as compared to the pre-pandemic period [ES = 41%; 95% CI: 32–50%] [Figure 8].

#### 3.7.3. Profession and Year of Training

Studies including only EM physicians [ES = 46%; 95% CI: 34–58%] had the highest prevalence of high DP as compared to studies including mixed EM workers [7 studies, ES = 40%; 95% CI: 30–51%] or ambulance crew [2 studies, ES = 39%; 95% CI: 37–41%] [Appendix A].

Noteworthy, among studies including mixed EM workers, high DP was highest among nurses [ES = 60%; 95% CI: 54–66%], and it was nearly equal between physicians [ES = 50%; 95% CI: 30–69%] and A/T staff [ES = 49%; 95% CI: 11–88%]. Meanwhile, paramedics had the lowest prevalence of high DP, accounting for 28% of the population [95% CI: 24–32%] [Appendix A].

Two studies reported the rate of high DP among EM residents. The meta-analysis revealed an increase in the prevalence of high DP as residents progressed from PGY-1 [ES = 65%; 95% CI: 61–68%] to PGY-2 [ES = 76%; 95% CI: 72–80%], which was equal to that of PGY-3 [ES = 76%; 95% CI: 72–79%] [Appendix A].

#### 3.7.4. Gender

Two studies reported the difference in high DP based on gender. The meta-analysis revealed no significant difference in the risk of developing high EE between males and females [logOR = 0.33; 95% CI: −0.04: 0.69; *I*^2^ = 33.26%, *p* = 0.22] [Appendix A].

### 3.8. Subgroup Analysis of Low PA

#### 3.8.1. Country

A total of 22 studies were included in this meta-analysis. Both Turkey and Iran showed the highest prevalence of low PA among EM workers [ES = 49%; 95% CI: 47–51%] and [ES = 49%; 95% CI: 43–56%]. Meanwhile, the US had the lowest prevalence of all [ES = 26%; 14–41%] [Appendix A].

#### 3.8.2. COVID-19 Pandemic

In this meta-analysis, 17 studies were conducted during the pre-pandemic period, while three studies were conducted during the COVID-19 pandemic. The prevalence of low PA doubled during the pandemic [ES = 62%; 95% CI: 32–88%] as compared to the pre-pandemic period [ES = 31%; 95% CI: 24–39%] [Figure 9].

#### 3.8.3. Profession and Year of Training

Studies including mixed EM workers [7 studies, ES = 38; 95% CI: 22–55%] had the highest prevalence of low PA as compared to studies including physicians only [11 studies, ES = 37%; 95% CI: 25–50%] or ambulance crew [2 studies, ES = 33%; 95% CI: 31–35%] [Appendix A].

Noteworthy, among studies including mixed EM workers, low PA was highest among nurses [ES = 55%; 95% CI: 49–61%], and it was nearly equal between physicians [ES = 43%; 95% CI: 32–53%] and A/T staff [ES = 40%; 95% CI: 33–46%]. Meanwhile, paramedics had the lowest prevalence of low PA, accounting for 35% of the population [95% CI: 30–40%] [Appendix A].

Two studies reported the rate of high DP among EM residents. The meta-analysis revealed an increase in the prevalence of low PA as residents progressed from PGY-1 [ES = 26%; 95% CI: 22–29%] to PGY-2 [ES = 34%; 95% CI: 30–38%] and PGY-3 [ES = 31%; 95% CI: 27–34%] [Figure 10].

#### 3.8.4. Gender

Two studies reported the difference in low PA based on gender. The meta-analysis revealed no significant difference in the risk of developing high EE between males and females [logOR = −0.11; 95% CI: −0.50: 0.27; *I*^2^ = 0%, *p* = 0.37] [Appendix A].

## 4. Discussion

This systematic review provides, by far, the greatest level of evidence regarding the global prevalence of burnout and its subdomains among healthcare workers in emergency care. Burnout is a prevalent problem in EM settings, ranging from 49.3% [22] to 58% [41], based on previous reports. A number of systematic reviews investigating the burden of burnout on EM personnel have been published in the past decade [42,43,44,45,46]. That being said, all of them were conducted prior to the occurrence of the COVID-19 pandemic. Additionally, all of them searched a small number of electronic databases which resulted in a limited number of studies, and therefore, their results are not generalizable. In our study, we included 29 studies (16,619 participants) which is the largest pool of EM populations across the literature. Our meta-analysis revealed an alarmingly high prevalence rate of burnout, high EE, high DP, and low PA of 43%, 39%, 43%, and 36%, respectively. Noteworthy, although 43% of the population suffered from burnout, regardless of the degree of severity, only 7% of participants were in burnout during the time of their interview. This supports our initial hypothesis that the prevalence of burnout, high EE or DP, and low PA surpass the rare event assumption (5%), and therefore, this problem is worthy of attention and addressed in clinical care. Our findings are quite different from those reported in the literature. In the systematic review of Adriaenssens et al. [42], burnout was reported in 26% of EM personnel. This difference could be attributed to the small sample size of the previous review (17 studies, 2021 participants) or to the fact that only nurses were analyzed, not the whole EM population (including physicians, A/T, paramedics, etc.).

Burnout can be attributed to numerous factors. High levels of emotional exhaustion (EE) are mainly attributed to factors such as overload of work, failure to balance work and personal life, fatigue, and the fear of making mistakes while practicing or second victim syndrome. High depersonalization (DA) levels occur due to an increased number of patients being subjected to hostile criticism by seniors and even patients, short temper, and sleep disturbance [2]. EM doctors also witness death almost daily and are burdened with the responsibility of relaying the information to friends and family of the deceased. Over time, a person becomes detached from almost all emotional feelings.

Beyond the prevalence of overall burnout, the present study highlights several key factors that could modify the prevalence of this phenomenon in this population. The occurrence of the COVID-19 pandemic has imposed a substantial psychological impact on the mental health of the world population, with a higher burden on EM personnel who were on the frontline against this disease [47]. This has been observed in our study, where we noted an increased rate of burnout and its subdomains during the pandemic, which doubled in the low PA domain as compared to the pre-pandemic period. This finding, although novel and has not been previously reported in the literature, warrants the attention of healthcare authorities in providing adequate measures to reduce the burden of burnout and its subsequent impact on EM workers’ intention to leave [48], quitting [49], and lowered medical care [50].

In our study, the prevalence of overall burnout, high EE and DP, and low PA were the highest among nurses, while physicians came in second. On the other hand, in the majority of cases, paramedics and A/T staff had the lowest rates of overall burnout, high EE, high DP, or low PA. This can be explained by the nature of nurses’ work, with demands them to do many tasks, spend a prolonged period of time with patients and their family members and friends [51], and be in greater proximity to traumatic cases and events (including workplace violence) [52]. That being said, our finding is inconsistent with that in the literature, where both physicians and nurses had similar rates of burnout across all of the study domains. For example, Zhang et al. [46] conducted a systematic review and included studies that only reported burnout among physicians. The authors reported a prevalence rate of high EE, high DP, and low PA of 40%, 41%, and 35%, respectively. Meanwhile, the review of Li et al. [43], who included nurses, only reported similar rates of 40.50%, 44.30%, and 42.70%, respectively. This difference in our observations could be related to the difference in the number of included studies and participants, along with the presence of substantial heterogeneity (>90%) in the previous two reviews as compared to ours.

In terms of geographical distribution, Singapore showed the highest prevalence of overall burnout, while the US had the highest prevalence of high DP, and Turkey/Iran had the highest prevalence of low PA. This finding is of great importance in directing the efforts needed to prevent and combat this problem among EM personnel. Additionally, gender-related differences in the prevalence of burnout have been previously reported in the literature, where women were more likely to experience burnout as compared to male peers [53]. In our study, we did not find any significant differences in the risk of experiencing overall burnout, high EE, high DP, or low PA among EM personnel based on gender. However, this finding is inconclusive secondary to the small number of analyzed studies in this subgroup (2 studies). Therefore, future studies are still needed to provide better evidence to either confirm or reject this observation.

## 5. Limitations and Future Directions

Although our study provides the biggest evidence, to date, regarding the worldwide prevalence of burnout among EM personnel, with special emphasis on the current COVID-19 pandemic, our research has several limitations. First, the number of included studies in certain subgroups (gender, year of training) was small, which further limits the applicability and generalizability of our findings. Second, the wide confidence interval reported in some of our findings limits the precision of our results, as the true effect estimate could lie anywhere between both ends of the spectrum. Third, several factors, like smoking [8] and marital status [8], have been reported to affect the risk of experiencing burnout among EM workers; however, due to the lack of enough data, we could not analyze these factors. Therefore, future studies are needed to focus on investigating more risk factors and effect-modifying variables to burnout and its subdomains. Finally, country-specific differences, which were noted particularly during the COVID-19 pandemic, could be related to the wide variability in the approach each country took in dealing with the pandemic, which might have resulted in an altered impact on healthcare workers’ work conditions, mental health, and burnout.

## 6. Conclusions

Burnout is a prevalent problem in emergency medicine practice, affecting all workers, including residents, physicians, nurses, paramedics, and administrative/technical staff. As residents progress through their training years, their likelihood of experiencing burnout intensifies. Nurses are most affected by this problem, followed by physicians. Country-associated differences in the rate of burnout and subdomains are evident. Organized efforts should be devoted to reducing the prevalence and associated burden of this problem. Given these findings, we advise the issuance of country-specific protocols that can address all of the mentioned issues in an attempt to lower the burden of burnout in healthcare settings, particularly at times of emergency, like the COVID-19 pandemic.

## Figures and Tables

**Figure 1 healthcare-11-02220-f001:**
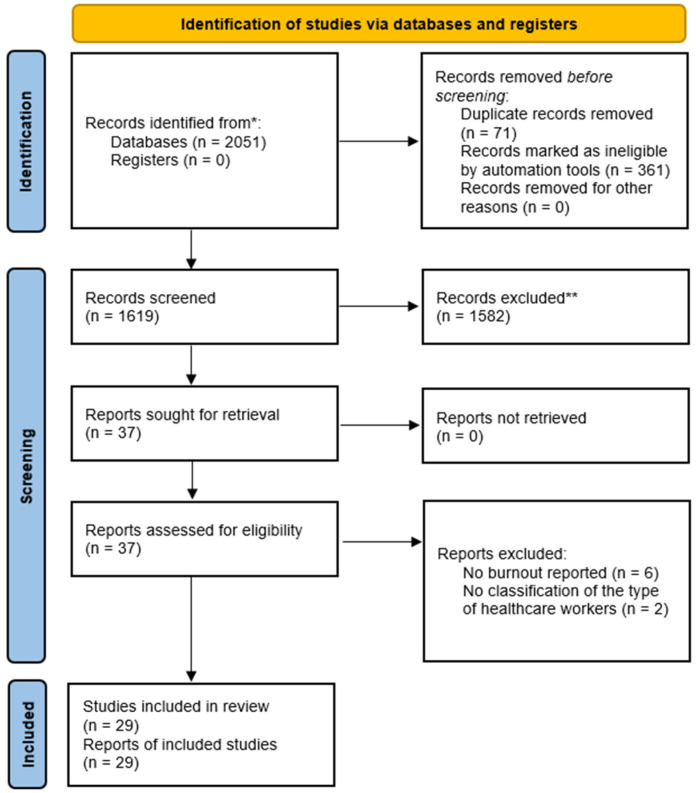
PRISMA flow diagram illustrating the database search and study selection processes. * records were identified through original database search; ** records were excluded based on title/abstract screening after the removal of duplicates through EndNote software (https://endnote.com/).

**Figure 2 healthcare-11-02220-f002:**
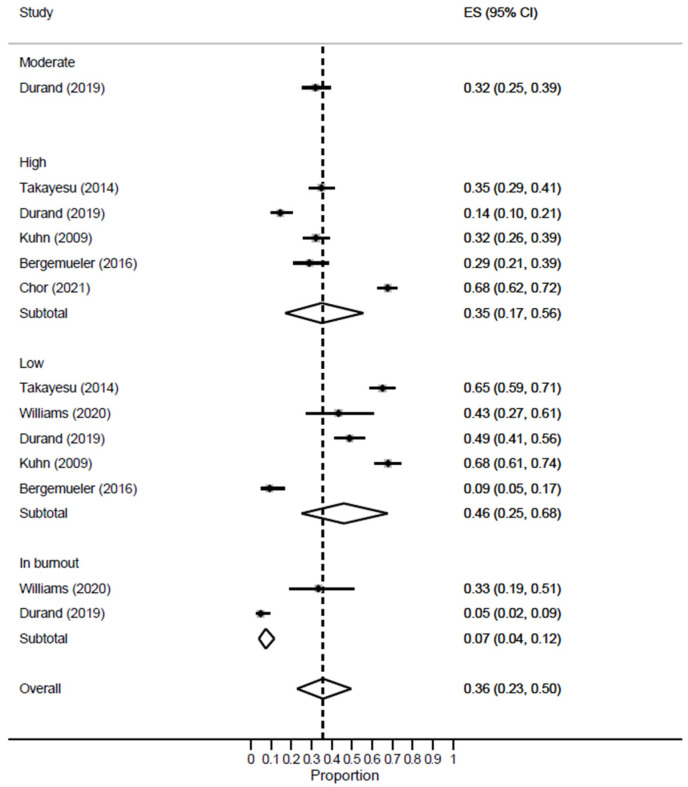
Forest plot of the prevalence of overall burnout among EM healthcare workers based on severity.

**Figure 3 healthcare-11-02220-f003:**
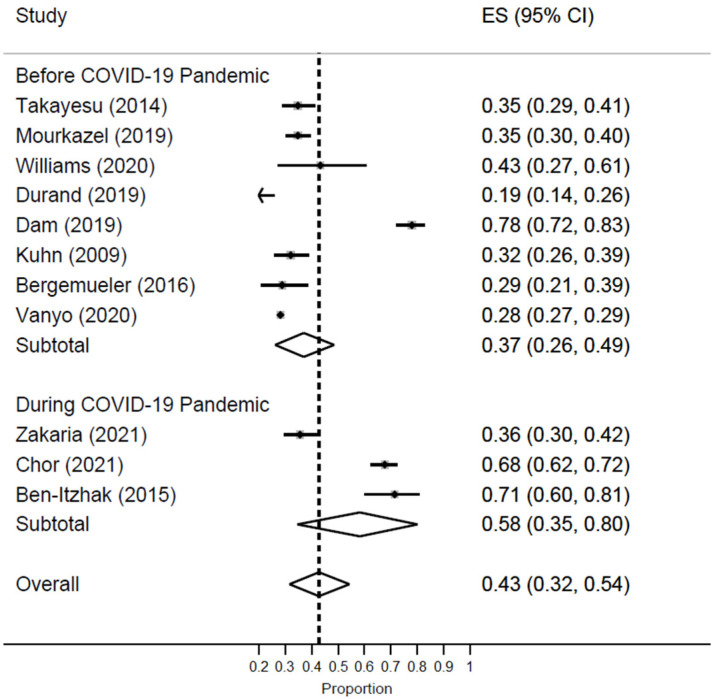
Forest plot of the prevalence of high burnout among EM healthcare workers based on COVID-19 pandemic status.

**Figure 4 healthcare-11-02220-f004:**
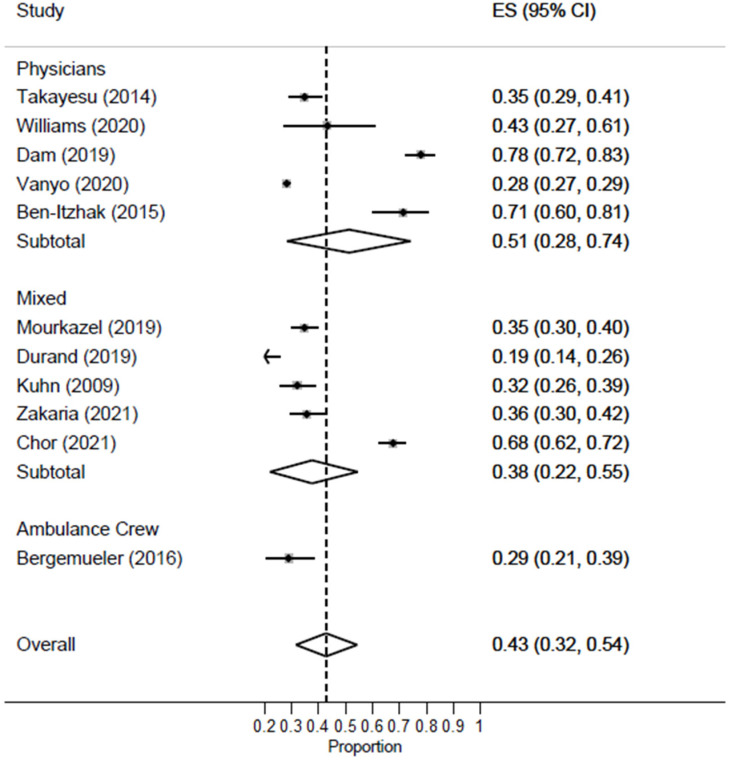
Forest plot of the prevalence of high burnout among EM healthcare workers based on profession.

**Figure 5 healthcare-11-02220-f005:**
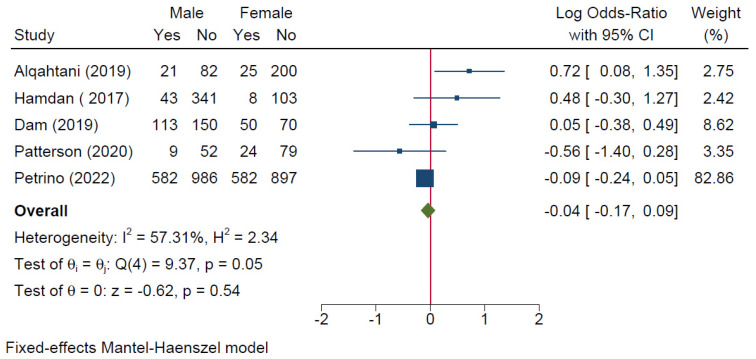
Forest plot of the risk of high burnout among EM healthcare workers based on gender.

**Figure 6 healthcare-11-02220-f006:**
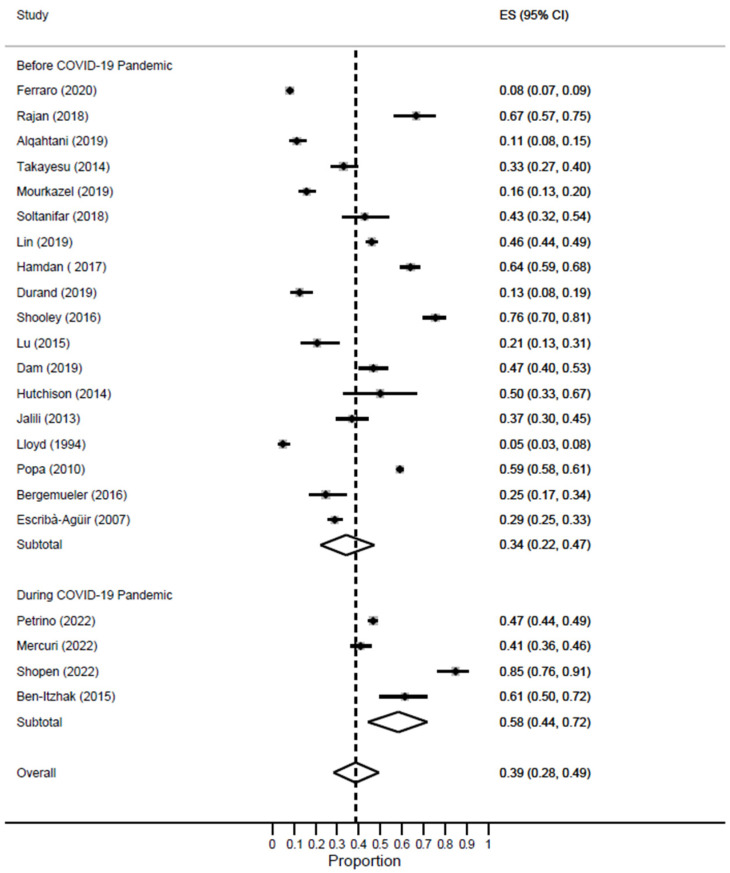
Forest plot of the prevalence of high EE among EM healthcare workers based on COVID-19 pandemic status.

**Figure 7 healthcare-11-02220-f007:**
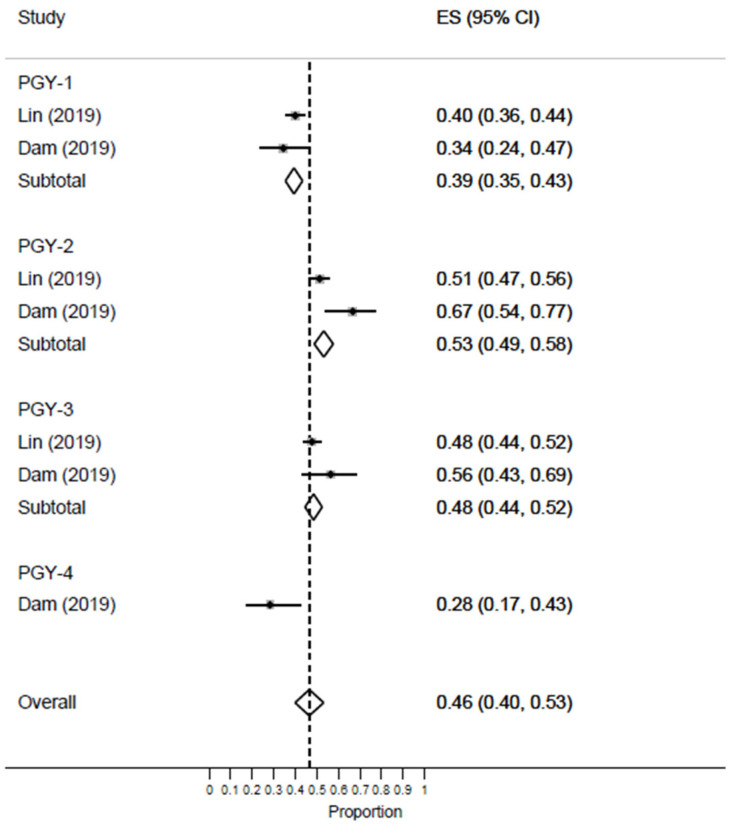
Forest plot of the prevalence of high EE among EM healthcare workers based on residents’ years of training.

**Figure 8 healthcare-11-02220-f008:**
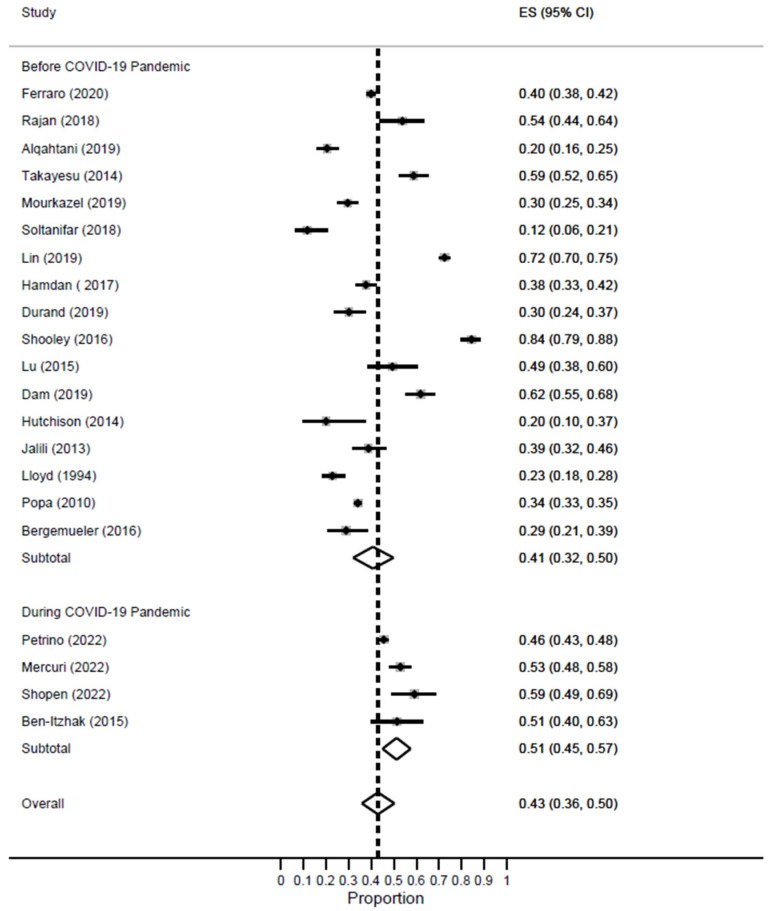
Forest plot of the prevalence of high DP among EM healthcare workers based on COVID-19 pandemic status.

**Figure 9 healthcare-11-02220-f009:**
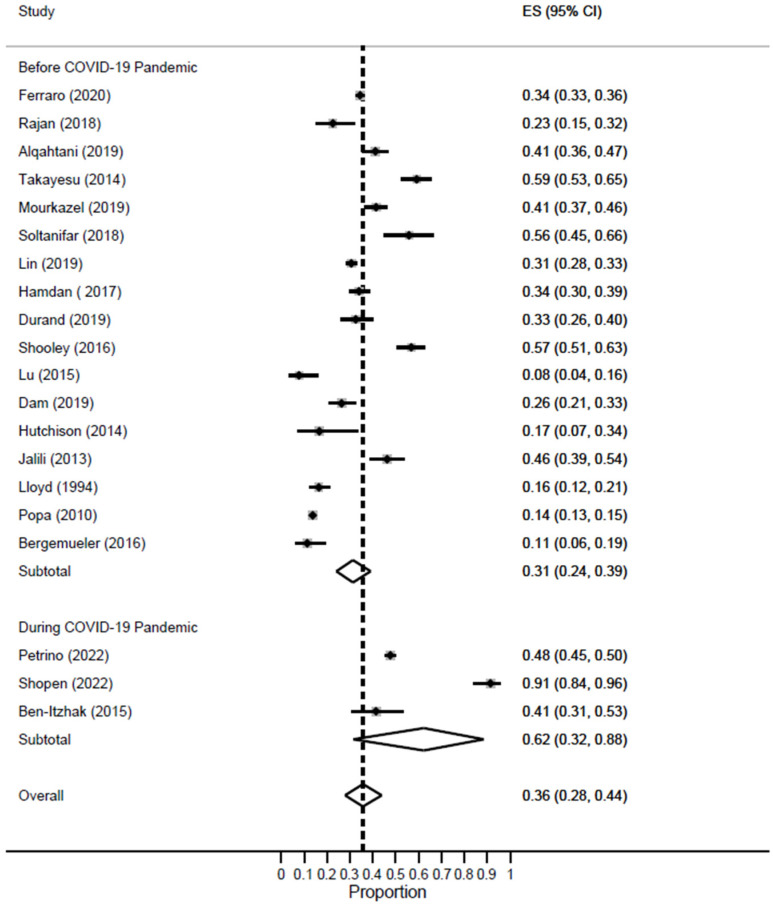
Forest plot of the prevalence of low PA among EM healthcare workers based on COVID-19 pandemic status.

**Figure 10 healthcare-11-02220-f010:**
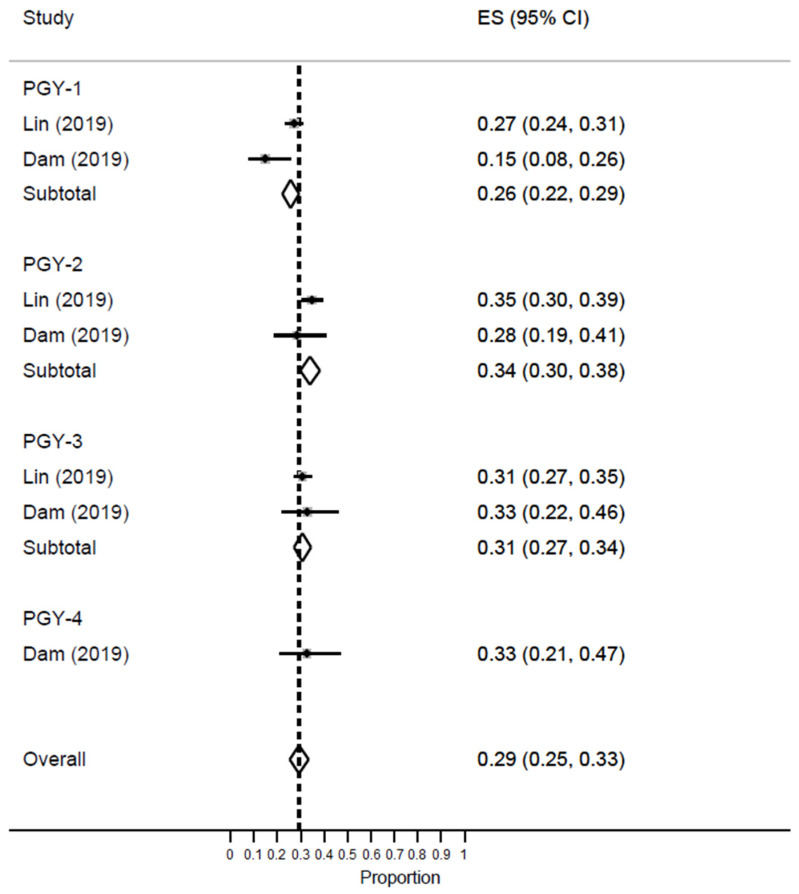
Forest plot of the prevalence of low PA among EM healthcare workers based on residents’ years of training.

**Table 1 healthcare-11-02220-t001:** Baseline characteristics of included studies and EM populations.

Author (YOP)	Study Design	Country	Population	Sample	Age (Mean)	Male	Female
Ferraro (2020) [11]	CS	Italy	Ambulance Crew	2361	34.8	2090	594
Rajan (2018) [36]	CS	South Africa	Physicians	100	35.1	42	51
Alqahtani (2019) [8]	CS	Saudi Arabia	Mixed	282	37	82	200
Takayesu (2014) [5]	CS	USA	Physicians	218	31.2	129	89
Mourkazel (2019) [32]	CS	France	Mixed	379	37.4	98	279
Soltanifar (2018) [39]	CS	Iran	Physicians	77	36	0	77
Lin (2019) [12]	CS	USA	Physicians	1522	31.3	879	643
Hamdan (2017) [26]	CS	Palestine	Mixed	444	32.5	341	103
Williams (2020) [40]	CS	USA	Physicians	30	32.1	-	-
Durand (2019) [10]	CS	France	Mixed	166	37.7	33	133
Schooley (2016) [37]	CS	Turkey	Mixed	250	-	135	115
Lu (2015) [31]	CS	USA	Physicians	77	-	48	29
Dam (2019) [23]	CS	USA	Physicians	222	31.8	150	70
Hutchison (2014) [27]	CS	Jamaica	Physicians	30	-	12	15
Jalili (2013) [28]	CS	Iran	Physicians	165	33.6	150	14
Lloyd (1994) [30]	CS	Canada	Physicians	268	38	233	35
Kuhn (2009) [29]	CS	USA	Mixed	193	-	140	53
Patterson (2020) [33]	CS	UK	Physicians	131	-	52	79
Popa (2010) [35]	CS	USA	Mixed	4725	33.6	3213	1512
Erdur (2015) [24]	CS	Turkey	Physicians	174	36.8	138	36
Bergemueler (2016) [20]	CS	Germany	Ambulance Crew	97	37	40	57
Escribà-Agüir (2007) [25]	CS	Spain	Mixed	639	-	356	279
Petrino (2022) [34]	CS	Turkey	Mixed	1925	-	1007	915
Mercuri (2022) [13]	CS	Canada	Physicians	416	44	220	196
Chang (2021) [9]	CS	USA	Physicians	22	-	-	-
Bhanja (2021) [21]	CS	USA	Mixed	944	39.3	221	421
Shopen (2022) [38]	CS	Israel	Physicians	88	-	59	29
Chor (2021) [22]	CS	Singapore	Mixed	337	-	109	228
Ben-Itzhak (2015) [19]	CS	Singapore	Physicians	337	-	49	21

YOP: year of publication; CS: cross-sectional; USA: United States of America.

**Table 2 healthcare-11-02220-t002:** The quality of included cross-sectional studies as per the Newcastle Ottawa Scale.

Author (YOP)	Selection	Comparability	Outcomes	Overall Rating
Ferraro (2020) [11]	4	2	2	Good
Rajan (2018) [36]	3	2	2	Good
Alqahtani (2019) [8]	4	2	2	Good
Takayesu (2014) [5]	3	0	2	Poor
Mourkazel (2019) [32]	3	2	2	Good
Soltanifar (2018) [39]	2	2	2	Fair
Lin (2019) [12]	3	2	2	Good
Hamdan (2017) [26]	3	2	2	Good
Williams (2020) [40]	3	0	2	Poor
Durand (2019) [10]	2	2	2	Fair
Schooley (2016) [37]	3	2	2	Good
Lu (2015) [31]	2	2	2	Fair
Dam (2019) [23]	3	2	2	Good
Hutchison (2014) [27]	3	2	2	Good
Jalili (2013) [28]	4	0	2	Poor
Lloyd (1994) [30]	3	2	2	Good
Kuhn (2009) [29]	3	2	2	Good
Patterson (2020) [33]	4	2	2	Good
Popa (2010) [35]	2	0	2	Poor
Erdur (2015) [24]	4	2	2	Good
Bergemueler (2016) [20]	3	0	2	Poor
Escribà-Agüir (2007) [25]	2	0	2	Poor
Petrino (2022) [34]	4	2	2	Good
Mercuri (2022) [13]	3	0	2	Poor
Chang (2021) [9]	4	2	2	Good
Bhanja (2021) [21]	2	2	2	Fair
Shopen (2022) [38]	3	2	2	Good
Chor (2021) [22]	2	2	2	Fair
Ben-Itzhak (2015) [19]	3	2	2	Good

YOP: year of publication. The selection domain was given an overall score of 4, and the comparability and outcomes domains were given overall scores of 2 and 3, respectively.

## Data Availability

Accessibility to the analyzed dataset can be granted upon reasonable request from the corresponding author.

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
