# Peer review of "The Global Prevalence and Associated Factors of Burnout among Emergency Department Healthcare Workers and the Impact of the COVID-19 Pandemic: A Systematic Review and Meta-Analysis"

_healthcare, 2023, doi:10.3390/healthcare11152220_

Round 1
Reviewer 1 Report
Thank you for the opportunity to read this paper. The paper opens up some potentially valuable international areas for research. Just one significant point and a couple of minor points.
First, the studies you employ are from different countries. These countries had different responses to covid. The approach a country took may have had an impact on burnout. This should at least be acknowledged as a limitation of the study even if not dealt with elsewhere in the article.
Second, in the discussion a reference to the assertion in the second sentence of the third paragraph would be helpful.
Third use of shortened terms like EE did make the paper difficult to read at times, although I appreciate why this was done.
Author Response
Comment 1: First, the studies you employ are from different countries. These countries had different responses to covid. The approach a country took may have had an impact on burnout. This should at least be acknowledged as a limitation of the study even if not dealt with elsewhere in the article.
Response: Thank you for highlighting this important point which is worthy of discussing. We added it to the limitations section as follows: Finally, country-specific differences which were noted particularly during COVID-19 pandemic could be related to the wide variability in the approach each country took in dealing with the pandemic, which might have resulted in an altered impact on healthcare workers’ work conditions, mental health, and burnout.
Comment 2: Second, in the discussion a reference to the assertion in the second sentence of the third paragraph would be helpful.
Response: Thank you. We added a reference supporting this statement (reference number 47).
Comment 3: Third, use of shortened terms like EE did make the paper difficult to read at times, although I appreciate why this was done.
Response: Thank you for highlighting this point. We acknowledge that the use of this term “EE” would be interrupting; however, we think that using it is better than the full meaning of the word as it will increase the word count of the manuscript because it is mentioned >80 times in the manuscript which is already 4980 words in number.
Reviewer 2 Report
The article raises an important problem in the group of health care workers.
The results can become the basis for creating prevention programs for employees.
The article, which is a review of the results of many studies, can therefore be the basis for developing a prevention program for employees of emergency departments. Currently, emergency department workers may experience the consequences of working during the pandemic, including burnout and psychosocial difficulties in functioning. I have no comments on the research methodology or references.
Author Response
Comment 1: The article raises an important problem in the group of health care workers.
The results can become the basis for creating prevention programs for employees.
The article, which is a review of the results of many studies, can therefore be the basis for developing a prevention program for employees of emergency departments. Currently, emergency department workers may experience the consequences of working during the pandemic, including burnout and psychosocial difficulties in functioning. I have no comments on the research methodology or references
Response: Thank you for your detailed feedback and positive review.
Reviewer 3 Report
This is a fine meta-analysis, worth publishing provided minor errors are corrected. See attached report for details.

Generally fine, except for minor errors mentioned in the attached report, and some undue "the" articles here and there (e.g. lines 308-309).
Author Response
Comment 1: Meta-analyses are very useful to develop a broad overview of a given problem. And since so much has been written on burnout among healthcare workers during the pandemic, this meta-analysis is more than welcome. It is well structured and quite elaborate. It is generally well written (but see comments below), the methodology is appropriate and well presented, and so are the results. The discussion is quite clear, and the authors are well aware of the limitations of their research. The conclusion, however, is a bit short, and could be expanded to include some suggestions for prevention.
Response: Thank you for your comment and detailed feedback. As for the conclusions part, we added the following sentence to provide suggestions for prevention: (lines: 471-473): Given these findings, we advise the issuance of country-specific protocols that can address all of the mentioned issues in an attempt to lower the burden of burnout in healthcare settings, particularly at times of emergency like the COVID-19 pandemic.
Comment 2: There are very few “format” errors in this paper; I have identified the following:
- Early on in the article, the authors mention Maslach on a number of occasions, but nowhere is she referenced in the text or in the bibliography.
Response: Thank you for highlighting this mistake. We added the missing citation, and now it’s reference number 3.
Comment 3: · Line 92: (“burnout” OR “depersonalization” OR…): missing quotation marks
Response: We appreciate your comment. However, we deliberately did not add quotation marks on these two terms. Upon doing database search, single-word keywords are added without quotation marks; however, keywords with two or more word should be added with quotation marks as to force the database to search for the exact keyword itself. For example, if we searched a database with the keyword “emergency medicine”, it will look for articles that reported the exact keyword “emergency medicine”. However, if we removed the quotation marks, the database will give us articles that used either word separately: either emergency or medicine.
Comment 4: · Line 308-308: “ the Turkey…”
Response: Thank you for your comment. We removed the word ‘the’.
Comment 5: · Line 412: should read “deceased” instead of “diseased”
Response: Thank you for highlighting this typo mistake. This has been corrected as suggested in the revised manuscript.
Comment 6: The problems mentioned above are minor and can be dealt with easily; otherwise, this article is quite good, so it is worth publishing after minor revision.
Response: We greatly appreciate your effort and time in reviewing and improving the quality of our manuscript.